# Near Infrared Spectroscopy for Prediction of Yeast and Mould Counts in Black Soldier Fly Larvae, Feed and Frass: A Proof of Concept

**DOI:** 10.3390/s23156946

**Published:** 2023-08-04

**Authors:** Shanmugam Alagappan, Anran Dong, Deirdre Mikkelsen, Louwrens C. Hoffman, Sandra Milena Olarte Mantilla, Peter James, Olympia Yarger, Daniel Cozzolino

**Affiliations:** 1Centre for Nutrition and Food Sciences, Queensland Alliance for Agriculture and Food Innovation (QAAFI), The University of Queensland, Brisbane, QLD 4072, Australia; 2Fight Food Waste Cooperative Research Centre, Wine Innovation Central Building Level 1, Waite Campus, Urrbrae, SA 5064, Australia; 3School of Agriculture and Food Sustainability, Faculty of Science, University of Queensland, Brisbane, QLD 4072, Australia; 4Department of Animal Sciences, University of Stellenbosch, Private Bag X1, Matieland, Stellenbosch 7602, South Africa; 5Centre for Animal Science, Queensland Alliance for Agriculture and Food Innovation (QAAFI), The University of Queensland, Brisbane, QLD 4072, Australia; 6Goterra, 14 Arnott Street, Hume, Canberra, ACT 2620, Australia

**Keywords:** black soldier fly larvae, NIR spectroscopy, yeast and moulds, feed quality, chemometrics

## Abstract

The use of black soldier fly larvae (BSFL) grown on different organic waste streams as a source of feed ingredient is becoming very popular in several regions across the globe. However, information about the easy-to-use methods to monitor the safety of BSFL is a major step limiting the commercialization of this source of protein. This study investigated the ability of near infrared (NIR) spectroscopy combined with chemometrics to predict yeast and mould counts (YMC) in the feed, larvae, and the residual frass. Partial least squares (PLS) regression was employed to predict the YMC in the feed, frass, and BSFL samples analyzed using NIR spectroscopy. The coefficient of determination in cross validation (R^2^_CV_) and the standard error in cross validation (SECV) obtained for the prediction of YMC for feed were (R^2^cv: 0.98 and SECV: 0.20), frass (R^2^cv: 0.81 and SECV: 0.90), larvae (R^2^cv: 0.91 and SECV: 0.27), and the combined set (R^2^cv: 0.74 and SECV: 0.82). However, the standard error of prediction (SEP) was considered moderate (range from 0.45 to 1.03). This study suggested that NIR spectroscopy could be utilized in commercial BSFL production facilities to monitor YMC in the feed and assist in the selection of suitable processing methods and control systems for either feed or larvae quality control.

## 1. Introduction

The world population is projected to grow up to 11 billion people by the end of this century, imposing a threat on global food security [1]. The growing population has also displayed changes in dietary patterns due to improved income [2,3]. These shifts in dietary habits and patterns have increased the global demand for protein, especially from animal sources. A recent report by the FAO has projected an increase (approx. 14%) in the demand and consumption of meat proteins by the year 2030 [4]. Patterns in consumption are also expected to increase up to 40% for poultry meat, followed by 34%, 20%, and 5% for pig meat, beef, and goat meat, respectively [4]. It has been recognized that livestock production by current practices accounts for about 18% of the total greenhouse gas emission derived from the agri-food production systems [5]. The production of meat protein using conventional feed ingredients is also challenged due to prevailing socio-environmental conditions, thereby suggesting the need for sustainable alternative feed sources [6,7,8].

The use of edible insects as a sustainable source of feed ingredient for livestock has been receiving increasing attention lately. The utilization of black solider fly larvae (BSFL), a saprophagous insect thriving in tropical and subtropical climatic conditions across the globe, has been of particular interest [9]. The BSFL undergoes six morphological stages or instars for a period of 10–52 days depending upon the feed source and rearing conditions, before they reach the pupal stage to become an adult fly [10].

Annually, 1.3 billion tons of the food generated for human consumption gets wasted [11]. Food waste constitutes for about 6% of the total anthropogenic greenhouse gas emissions [12]. The use of multi-stream food waste as substrate for rearing BSFL is becoming increasingly popular in commercial production facilities. BSFL reared on such food waste streams are rich in several nutrients including crude protein (range 32–46%), crude fat (range 15–38%), fibre (range 20.4–30.5%), and micronutrients [13,14,15,16]. Recent studies have also revealed that the replacement of conventional feed ingredients with BSFL meals in broilers, pigs, and aquaculture has positively influenced several growth performance parameters [17,18,19] making them a sustainable alternative to conventional feedstuff.

Feedstuffs and feed ingredients intended to be used as animal feed must be monitored for several microbial risks and regulated according to several legislations polices [20,21]. Yeast and mould counts (YMCs) are a combination of microbial organisms that are monitored in animal feeds in several regions across the globe [20,21]. The presence of these microorganisms can affect, among other things, the palatability of the feedstuff, growth, and the performance of domestic animals. It is also well established that several species of mould can produce mycotoxins, a secondary metabolite that can have adverse health effects in animals upon ingestion [22]. Previous studies have reported that BSFL reared on homogenous side stream wastes might vary in YMC between 2.0 to 7.0 log CFU/g depending upon the feed source and post-harvest conditions [23,24,25]. Therefore, it will be of importance for the feed industry to monitor YMC in BSFL when reared with different complex organic waste streams.

The detection and quantification of yeast and mould by conventional methods can be time consuming and laborious (e.g., culture of the microorganism in selective agars, microscopy, serology, histopathology, molecular markers, among others). On the other hand, the development of rapid and non-invasive methods will enable commercial BSFL production facilities to monitor microbial load as well as to assist in the implementation of a HACCP system or other feed safe protocols required by the industry [26]. Over the years, near infrared (NIR) spectroscopy combined with chemometric techniques has been showcased due to its potential to monitor product quality and ensure food safety in different fields [27,28,29,30]. NIR spectroscopy combined with chemometric techniques has been employed to determine the total viable counts and other microbial parameters in a diverse set of food samples (e.g., beef and pig meat, and fish) [31,32,33,34,35]. However, to date no research has been reported on the utilization of NIR spectroscopy to predict yeast or mould counts in BSFL.

The aim of this study is to evaluate the ability of NIR spectroscopy in combination with chemometric techniques to determine the YMC of feed, frass, untreated, and processed larvae reared with different food waste streams.

## 2. Materials and Methods

### 2.1. Sample Preparation

Five different feeds (2 homogenous/single food waste streams; 3 complex/heterogenous food waste streams) were used for the rearing of BSFL. The experiment was carried out using a homogeneous waste stream comprised of soy waste (SW) and customized bread-vegetable (BV) and the rearing conditions defined by the commercial site “A” were followed. The experiments carried out using the complex food waste streams included supermarket waste (food waste primarily comprising of bread, packed pastries, bottled milk, and chips) childcare center waste (primary constituents include salads, sandwiches, and cooked food) and a food waste mixture (mixture of waste from supermarkets, retail stores, childcare centres, and fast-food restaurants). The packaging material from these food wastes were removed and was used as feed for BSFL where the rearing conditions defined by the commercial site “B” were followed.

#### 2.1.1. Experiment 1—Rearing of BSFL Using Homogenous Waste

The homogenous SW obtained from a custard manufacturer was utilized as feed for rearing BSFL. Customized BV waste was prepared by creating a mixture containing bread 60% (*w*/*w*), and a combination of peaches, potatoes, carrots, apples, and bananas (8% each *w*/*w*) purchased from a local supermarket. Two kg of SW was added to 6 trays (dimensions 60 × 40 × 12 cm) whilst the remaining feed required throughout the experiment was stored at 4 °C. A feed subsample was collected and stored at −20 °C prior to NIR analysis. A fraction of this feed was suspended in 40% (*v*/*v*) glycerol at a 1:1 (*w*/*v*) ratio and stored at −80 °C for YMC analysis.

Two kilograms of the starter feed from the BV experiment were prepared by chopping 1.2 kg of bread and 160 g of each of the fruits and vegetables. The chopped bread and vegetables were mixed and added to 6 trays (60 × 40 × 12 cm). Unlike SW, the feed utilized in this experiment was prepared fresh, where a subsample (approx. 20 g) was collected each time the BSFL in the trays were being fed and stored at −20 °C. The stored BV feed samples were pooled together before NIR analysis and stored at −20 °C. A sub-sample was suspended in 40% (*v*/*v*) glycerol at a 1:1 (*w*/*v*) ratio and stored at −80 °C for YMC analysis.

Approximately 10,000 5-day old (chronological age) BSFL obtained from hatched eggs were added into each of the 6 trays containing the starter feed (SW and BV feed). The trays were then subjected to the rearing conditions established by the industrial facility and fed ad libitum. The trays for both experiments were prepared for harvesting the 5th instar BSFL upon sighting the 1st pre-pupae. The larvae were picked using sterile tweezers from random spots in each of the trays (n = 6 trays for each waste type). The frass available upon harvesting the 5th instar larvae was collected from random spots in each of these trays. The 5th instar larvae and 5th instar frass samples were stored at −20 °C prior to NIR analysis where a fraction of these samples was suspended in 40% (*v*/*v*) glycerol at a 1:1 (*w*/*v*) ratio and stored at −80 °C for YMC analysis. The 6th instar larvae and 6th instar frass were harvested following the same protocol described above upon sighting of the first pupae and, stored at −20 °C prior to NIR analysis where a small sample fraction was suspended in 40% glycerol as stated above and stored at −80 °C for YMC analysis.

#### 2.1.2. Experiment 2—Rearing of BSFL Using Heterogenous Waste

The three heterogenous food waste streams: supermarket waste, childcare centre waste, and a food waste mixture as described in Section 2.1 were received from local suppliers, de-packaged, and grinded using an industrial grinder. Three kg of the grinded waste from each of the waste streams were placed into trays (n = 3 for each waste, 60 × 40 × 12 cm) where approximately 15,000 of 5-day old BSFL (chronological age) were added to each tray. The trays were then subjected to the rearing conditions established by the industrial facility B and fed ad libitum. The feed for these experiments was de-packaged and grinded on the day of feeding where a subsample (approx. 20 g) from each of the waste streams was collected, pooled, and stored at −20 °C prior to NIR analysis. A fraction of the stored feed samples from the three different waste streams was suspended in 40% (*v*/*v*) glycerol at a 1:1 (*w*/*v*) ratio and stored at −80 °C for yeast and mould analysis. The 5th instar BSFL and frass from these trials were harvested and stored for analysis as mentioned above in Section 2.1.1. It is to be noted that 6th instar BSFL and frass were not collected in these experiments.

The temperature and other production conditions were monitored by each of the facilities. The feeding, waste management system, and rearing conditions are protected by industrial rights. Overall, this study has used industrial operating procedures and standards that can be used by similar types of industries worldwide.

#### 2.1.3. Post-Harvest Processing of BSFL

The BSFL after harvesting was subjected to three different processing treatments. The whole live larvae samples were homogenized using a food processor (Cuisinart Food Processor, Sydney, NSW, Australia). The second treatment involved blanching the live larvae at 100 °C for 5 min and homogenizing them using a food processor (Cuisinart Food Processor, Australia). The final treatment involved blanching live larvae at the above-mentioned conditions, accompanied with drying using a commercial dehydrator at 70 °C for 12 h followed by homogenization with the food processor (Cuisinart Food Processor, Australia). After processing, the samples were stored at −20 °C prior to NIR analysis. A sub-sample of this was suspended in 40% (*v*/*v*) glycerol at a 1:1 (*w*/*v*) ratio and stored at −80 °C for YMC analysis. It is to be noted that blanching and drying treatments were carried out only on BSFL samples from Facility B.

### 2.2. Determination of Yeast and Mould

Samples (feed, larvae, and frass) collected from the five different waste streams described in the previous sections were analyzed using routine mould and yeast reference methods. The Australian standard: AS 5013.29-2009 was followed with some modifications [36]. In brief, 0.1 g of sample was suspended in 40% glycerol broth and homogenized with 0.9 mL peptone water (0.1%, *w*/*v*, Merck, Rahway, NJ, USA) and serial diluted to appropriated concentrations with the same diluent. Samples with water activity (aw) > 0.95 were plated onto Dichloran Rose Bengal Chloramphenicol (DRBC, Merck) agar plates where samples with aw < 0.95 were plated onto Dichloran 18% Glycerol (DG18, Merck) agar plates. The Aw of each sample was measured using a Novasina LabSwift-AW water activity meter (Novasina AG, Lachen, Switzerland). All plates were aerobically incubated with the lid in uppermost position at 25 °C for 5 days before enumeration. The *Penicillium commune* (laboratory strain) and *Saccharomyces cerevisiae* (laboratory strain) were streaked on both DRBC and DG18 agar plates and used as positive controls. Both plates were incubated under the same incubation conditions as described above. All the analysis was carried out in biological triplicates.

### 2.3. NIR Analysis

Stored samples were thawed at room temperature and analyzed using a Fourier Transform-NIR instrument (Tango-R, Bruker Optics GmbH, Ettlingen, Germany). Samples were placed in a glass cuvette (10 mm diameter) where the spectra were collected as the average of 64 interferograms at a resolution of 4 cm^−1^ in the wavenumber range of 11,550 to 3950 cm^−1^ (OPUS software, version 8.5, Bruker Optics GmbH, Ettlingen, Germany). The cuvettes were cleaned with 70% ethanol and wiped dry using paper towels between samples. The NIR raw spectra, baseline corrected spectra, first (1, 10, 10, 1), and second (2, 10, 10, 2) derivatives are shown in the Appendix A.

### 2.4. Data Analysis

The Vektor Direktor (version 1.0, KAX Group, Sydney, NSW, Australia) was used for multivariate data analysis. The NIR spectral data were smoothed and pre-processed using the Savitzky–Golay second derivative (second order polynomial and a smoothing window size of 10 points) before analysis. Trends, patterns, and outliers in the data set were visualized by performing principal component analysis (PCA) [37]. Partial least squares regression (PLS) [38] was utilized to develop models for the prediction of YMC in the feed, frass, and BSFL samples obtained from the different experiments [39]. A summary of the different data sets (number of samples) used to develop the PLS models is depicted in Table 1. To evaluate the performance of the PLS models, the original dataset was split into two subsets, namely calibration and validation, using the Kennard-Stone algorithm provided by the software (Vektor Direktor, version 1.0, KAX Group, Sydney, NSW, Australia). In this study, approximately 70% of the samples were selected and used to develop the cross-validation models, while 30% of the samples were used for validation. By performing data partitioning, knowledge of the training dataset did not affect the test dataset, and the predictive power of the created model subsequently increased. A leave-one-out cross validation was applied during the development of the PCA and PLS models. The cross-validation models were evaluated using the coefficient of determination (R^2^_CV_), the standard error in cross validation (SECV), bias, slope, and the residual predictive deviation (RPD) (SD/SECV) [40,41]. The validation models were evaluated using the standard error in prediction [40,41].

## 3. Results and Discussion

### 3.1. Yeast and Mould Counts

The average and standard deviation of the YMC determined in each of the samples is shown in Figure 1. It can be observed that the moulds were distributed at higher concentrations in frass than in larvae and feed. It has been reported that BSFL gut is characterized with a diverse set of fungal communities including *Candida*, *Dipodascaceae*, *Trichosporon*, *Pichia*, species where their occurrence depends upon the substrate used for rearing the larvae [42,43,44]. Recent studies have also reported the passage of different genera of bacteria present in the guts of BSFL through their digestive tracts as consequence of the digestion of different waste streams or feeds used to grow the larvae [45]. Therefore, it can be suggested that fungal communities could also occur in the residual frass after the digestion process. However, the fungal count in the supermarket waste frass was found to be lower than that in the waste source. This can possibly be accounted for due to the relatively low water activity (0.85) observed in the supermarket frass compared to the other frass samples (>0.91).

In this study, the YMC in the frass obtained from the 6th instars samples from site A was higher than the frass obtained from the 5th instar (Figure 1). It has been reported that BSFL upon reaching 6th instar (pre-pupae) will stop feeding on the substrate, empty their digestive tract and crawl off to a dry clean site to commence pupation [46]. The emptying of the digestive tract during the pre-pupal stage could explain the high counts in the frass obtained from the 6th instar frass compared to the 5th instar. Furthermore, the 6th instar BSFL, despite having emptied their guts, was found to have higher YMC when compared to the 5th instar larvae when reared with both SW and BV waste. It is to be noted that the larvae samples in these experiments were not surface sterilized upon harvest. Therefore, it is suggested that the high counts observed in 6th instar larvae is probably due to cross-contamination of the exterior of the larvae that is in contact with the frass.

The YMC for the larvae reared on heterogenous waste streams was observed to be low when compared to that of BSFL reared on homogenous waste streams (SW and BV waste). It is well established that the differences in the rearing substrate and rearing conditions modulate the nutritional composition of the larvae and microbial counts [47,48,49]. The BSFL has also been reported to exert antimicrobial action against pathogenic microbes based on the feed source and rearing conditions employed [50,51,52].

It can be also seen from Figure 1 that the YMC of BSFL larvae subjected to processing, such as blanching and drying, were reduced to 1.3 log CFU/g. Campbell and collaborators [23] reported similar levels of reduction in YMC upon subjecting the BSFL to two different thermal processing treatments. The ability of thermal processing to reduce the microbial load of several pathogenic microorganisms in BSFL has been demonstrated in previous studies [53,54].

### 3.2. Principal Component Analysis

A principal component analysis (PCA) was performed using the NIR spectra collected from all the experiments (n = 75). The first two principal components (PC) explain 88.34% of variance in the samples. It can be observed from the PCA scores plot (Figure 2) that the frass and larvae samples from the different rearing experiments were well separated and formed separate clusters unlike the feed samples which exhibited limited separation. It is also noteworthy that the BSFL samples subjected to the drying process were found to be clustered and separated from the rest of the larvae samples. The PCA loading plot showed that the highest loadings in PC1 had characteristic bands in the region around 7200 cm^−1^ and 5300 cm^−1^ which is associated with the water content [55,56] (Figure 3). The bands observed in the region around 5800 cm^−1^ and between 4900 cm^−1^ to 4600 cm^−1^ could be attributed to lipid and protein content, respectively [57,58,59]. The highest loadings in PC2 exhibited positional shifts in the bands around 7100 cm^−1^ and 5300 cm^−1^ which could be associated with changes in water activity of the sample. Similar positional shifts were observed in regions around 4900 cm^−1^ and 4600 cm^−1^ (Figure 3). These shifts could be attributed to the difference in physical characteristics of the samples and explained by processing and other factors (e.g., site, waste stream, etc.) [26].

### 3.3. Descriptive and Cross Validation Statistics

The descriptive statistics (mean, standard deviation, range, and CV) for the YMC measured in the different data sets used to develop the PLS models are shown in Table 2. The mean YMC values range from 2.1 log CFU/g in the larvae samples to 4.6 log CFU/g in the frass samples. The CV for the YMC values showed that the complete data set has the highest variability where the lowest was found to be in the frass samples. Overall, the descriptive statistics indicated that there was good dispersion in the samples to attempt PLS calibration for YMC using NIR spectroscopy.

The cross-validation and prediction statistics for the measurement of YMC in the different data sets (larvae, feed, and frass) from the different waste streams and sites are reported in Table 3. The coefficient of determination in cross validation (R^2^_CV_) and the standard error in cross validation (SECV) obtained for the prediction of YMC were for feed (R^2^_cv_: 0.98 and SECV: 0.20), frass (R^2^_cv_: 0.81 and SECV: 0.90), larvae (R^2^_cv_: 0.91 and SECV: 0.27). The RPD value (4) obtained for the prediction of YMC in the feed suggested that this model could be used to predict YM counts in the feed sources intended to be used for feeding the BSFL. The RPD values for the prediction of YMC in the frass and larvae were 3.1 and 4, respectively. The cross-validation statistics using samples from facility A were R^2^cv: 0.90, SECV: 0.45, and RPD: 3.5 while for facility B they were R^2^cv: 0.80, SECV: 0.67, and RPD: 2. These cross-validation models were considered moderately accurate, suggesting that they could be used for screening the samples for the occurrence of YMC in the samples. The cross-validation statistics for the models developed using all samples (combined set) were R^2^cv: 0.74, SECV: 0.82, and RPD: 2.1. The predictive ability of the cross-validation models was evaluated using the validation set created by applying the Kennard-Stone algorithm explained in the Section 2. Moderate predictions of YMC in all the samples were obtained as indicated by the moderate standard error of prediction (SEP) obtained (range 0.45 to 1.03). The predicted vs. reference values plot for the different cross-validation models are displayed in Figure 4A–F. The number of latent variables (LV) used to develop the cross-validation models were 2 for feed and frass, 5 for facility A and the combined set, 6 LV were used to develop the models using the samples from facility B, and 8 for larvae.

Previous results from our group indicated that BSFL can be cluster based on it its morphological stage and the feed source used to rear the larvae [26]. It was discussed in the previous section that the feed, frass, and the larvae samples employed in this study were clustered in groups, suggesting spectral differences among the samples. It can also be suggested that the poor cross-validation statistics obtained for some of the models developed could be related to some of the limitations of the reference method. Furthermore, some of the data sets have shown a very low concentration of YMC.

The loadings plot for the PLS models used to predict YMC in the feed, larvae, and frass samples are reported in Figure 5A–C. The loadings derived from the PLS regression models were found to be different with few similarities between the feed and frass samples analyzed. The highest and most similar loadings for the prediction of YMC in the feed and frass samples were observed in the region around 5100 cm^−1^ which is associated with O-H stretching, HOH, and N-H bending combinations that can be associated with water and proteins, whilst that around 5800 cm^−1^ is associated with asymmetric and symmetric stretching of C-H in CH_2_ in the first overtone region, which is mainly related to lipids. The bands around 7100 cm^−1^ are associated with O-H stretching and bending vibrations in the first overtone attributed to the water content in the samples [55,56] and, the band around 8700 cm^−1^ is due to the C-H vibrations in the second overtone region that can be linked to either lipids or aromatic groups [57,60].

In addition, the feed samples showed the highest loadings around 5300 cm^−1^ which is associated with moisture content of the samples, and around 5968 cm^−1^ which is linked with C-H aromatic groups while around 6928 cm^−1^ with N-H asymmetric vibrations in the 1st overtone region—which are attributed to proteins [55,56]. The bands around 5968 cm^−1^ and 6928 cm^−1^ are also associated with C-H (2υ) vibrations in aromatic groups [58,60]. The highest loadings for the prediction of YMC using the frass samples were observed around 4352 cm^−1^ and 4784 cm^−1^, associated with C-H bending (aromatic groups) and O-H, respectively [58,59,61] whilst the band around 6032 cm^−1^ corresponds with C-H vibrations in the 1st overtone region attributed to lipids [59,60]. The highest loadings used to predict YMC using the larvae samples were observed around the regions 4848 cm^−1^, 5088 cm^−1^, 5952 cm^−1^, and 6800 cm^−1^. The band at 4848 cm^−1^ is associated with C-H bending (aromatic groups), around 5952 cm^−1^ with C-H (2υ) vibrations in the aromatic groups, while around 6800 cm^−1^ with protein [57,58,60]. The band at 5088 cm^−1^ is associated with O-H and C-H combinations related to polysaccharides [60].

## 4. Conclusions

The findings of this study indicated that NIR spectroscopy combined with PLS regression was able to predict YMC with good and moderate accuracy in the feed samples and residual frass. The cross-validation statistics for the prediction of YMC in the larvae, despite exhibiting different spectral characteristics, did not yield prediction models suitable for screening fungal contamination. The reference data imposed certain restraints that influenced the accuracy of the PLS model developed for the larvae samples. The use of NIR spectroscopy in combination with chemometrics can be used as an instantaneous tool to monitor the quality of the feed samples intended to be used as substrate for rearing BSFL.

Overall, this study suggested that NIR spectroscopy could be utilized in commercial BSFL production facilities to monitor YMC in the feed, assist in the selection of suitable processing methods to be applied based on the extent of contamination observed in the substrate, and implement control systems either for feed or larvae quality control.

## Figures and Tables

**Figure 1 sensors-23-06946-f001:**
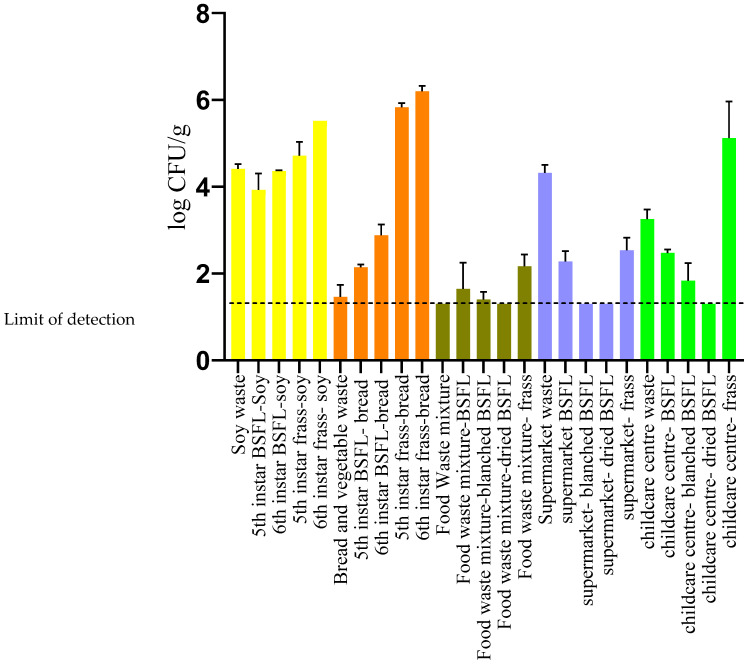
Yeast and mould count of BSFL, feed, and frass samples employed in the study.

**Figure 2 sensors-23-06946-f002:**
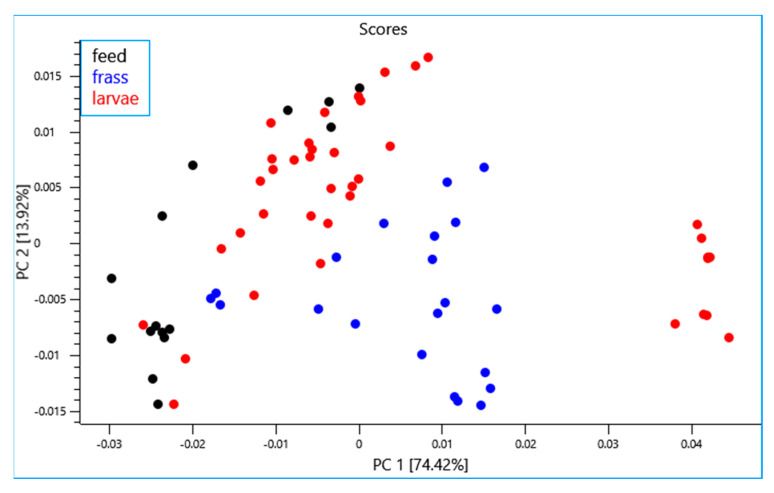
Principal component score plot of all samples (data sets) analyzed using near infrared reflectance spectroscopy.

**Figure 3 sensors-23-06946-f003:**
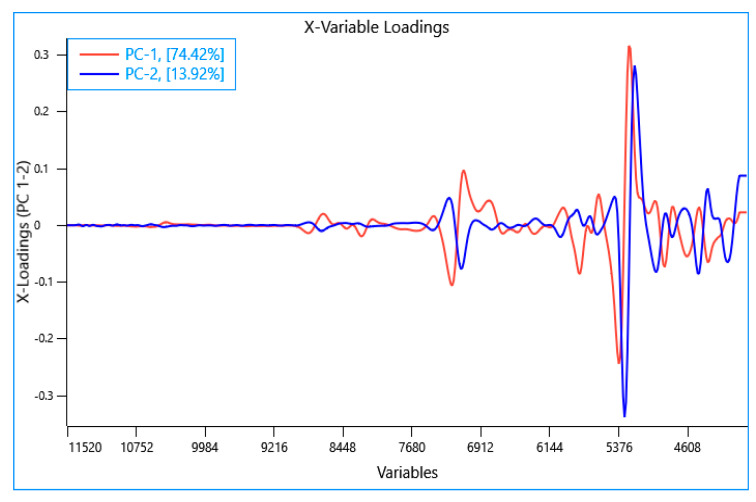
Principal component loadings plot of all samples (data sets) analyzed using near infrared reflectance spectroscopy.

**Figure 4 sensors-23-06946-f004:**
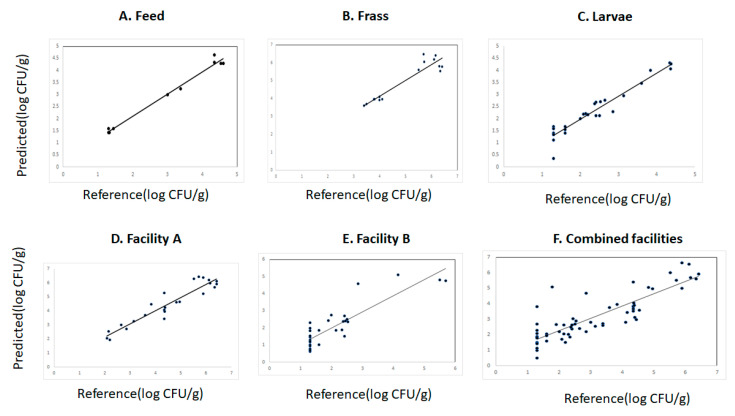
Predicted vs. reference plot for the prediction of yeast and mould counts in the different data sets analyzed using near infrared reflectance spectroscopy.

**Figure 5 sensors-23-06946-f005:**
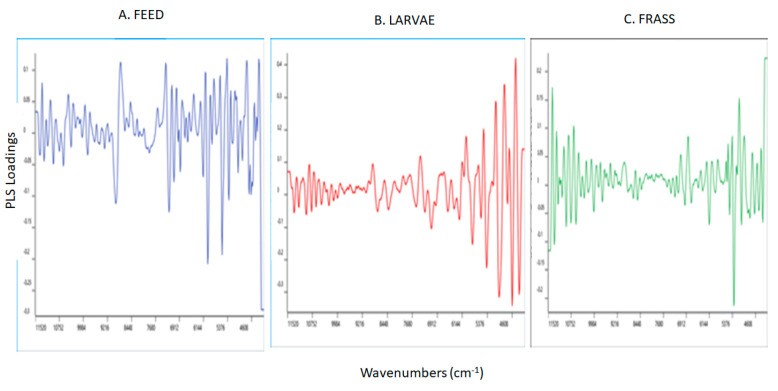
Partial least squares loadings for the optimal models used to predict yeast and mould counts in the feed, larvae, and frass samples analyzed using near infrared reflectance spectroscopy.

**Table 1 sensors-23-06946-t001:** Summary of different datasets used to develop the partial least square regression models.

Data Set	Details of Samples
Feed (n = 15)	Substrate used for rearing BSFL: SW, customised BV, supermarket waste, childcare waste, waste mixture.
Larvae (n = 39)	5th and 6th instar BSFL from SW and BV waste; unprocessed, blanched, and dried larvae from supermarket waste, childcare waste, waste mixture.
Frass (n = 21)	5 and 6th instar frass from soy and BV waste trials. 5th instar frass from supermarket waste, childcare waste, waste mixture
Facility A (n = 30)	Feed, 5th and 6th instar BSFL and frass from soy waste and BV waste. Homogenous data set.
Facility B (n = 45)	Feed, unprocessed, blanched, dried BSFL, and frass from supermarket waste, childcare waste, waste mixture.
Both facilities together (n = 75)	Facility A + B

**Table 2 sensors-23-06946-t002:** Descriptive statistics for the yeast and mould count in the different datasets used to develop the partial least squares regression analyzed using near infrared spectroscopy.

Samples Set	N	Mean	SD	Max	Min	CV%
Feed	15	2.9	1.4	4.6	1.4	48.2
Larvae	39	2.1	1.1	4.4	1.4	52.3
Frass	21	4.6	1.6	6.5	2	34.7
Facility A	30	4.2	1.6	6.5	1.4	38.1
Facility B	45	2.2	1.2	5.7	1.4	54.5
All samples (feed, frass and facilities)	75	3	1.7	6.5	1.4	56.6

N: number of samples, SD: standard deviation, max: maximum, min: minimum, CV: coefficient of variation (SD/mean × 100).

**Table 3 sensors-23-06946-t003:** Cross validation and prediction statistics for the measurement of yeast and mould count in the different data sets analyzed using near infrared reflectance spectroscopy. N: number of samples, R^2^_cv_: coefficient of determination in cross validation, SECV: standard error in cross validation, RPD: residual predictive deviation (SD/SECV), LV: latent variables, SEP: standard error of prediction.

	n	R^2^_cv_	SECV	Bias	Slope	LV	SEP
Feed	10	0.98	0.20	0.042	0.93	2	0.45
Frass	18	0.81	0.90	0.13	0.80	2	0.98
Larvae	30	0.91	0.27	−0.029	0.95	8	0.65
Facility A	20	0.90	0.45	−0.006	0.95	5	0.67
Facility B	30	0.80	0.62	−0.007	0.93	6	0.97
All samples (feed, frass, and facilities)	65	0.74	0.82	−0.025	0.80	5	1.03

## Data Availability

Not Applicable.

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
