# Peer review of "Near Infrared Spectroscopy for Prediction of Yeast and Mould Counts in Black Soldier Fly Larvae, Feed and Frass: A Proof of Concept"

_sensors, 2023, doi:10.3390/s23156946_

Round 1
Reviewer 1 Report
The manuscript contains the results of solid a well presented research on an important problem. It can be published after taking the following minor and major comments into account.
No need to indicate both R2 and RPD values in the abstract, because they are strictly related and can be recalculated to each other. Therefore, this is a redundant information which is unwanted. Report one of them in the abstract or tables. Of course, both can be used in the text.
Any classical PLS reference is missing. Please add at the first mentioning.
Table 1 and 2 should be better explained. Was the total number of samples 75? It does not coincide with the total sum of N. Probably feed+larvae+frass=75 and facility A+B=75, but it is not evident from the table
The raw NIR spectra of all samples should be provided to estimate their quality and difference between spectra. Also, the main peaks assignment made in the discussion part cannot be really followed using the loadings of the derivative spectra only.
The number of latent variables LVs used for the PLS model building is has not been discussed implicitly. As it indirectly follows from Figure 5, the optimal models used to calculate the validation statistics were: 5, 5, and 10 for feed, BSFL and frass, respectively. While LV = 5 looks reasonably, LV = 10 is inapplicable considering the number of samples and design used. Also, the system physical complexity is approximately the same for all sample groups, and therefore, the numbers of LVs should be similar.
In fact, what is missing in the manuscript is the thorough discussion of the model complexity using different approaches, RMSEC(CV) vs nLV, loading shapes etc. Considering the system complexity and multiple error sources, no perfect statistics is expected for these PLS models, but realistic values should be provided for reliable comparison and discussion.
The language quality is high. Only minor improvements and proofreading are required.
Reviewer 2 Report
The manuscript entitled “Near Infrared Spectroscopy for prediction of yeast and mould counts in black soldier fly larvae, feed and frass: a proof of concept” written by Alagappan S. et. all presents the results of using the NIR spectroscopy coupled with chemometric analysis (PLS) to facilities monitor yeast and mould counts in the feed, larvae, and the residual frass of black soldier fly. This manuscript can be published after minor changes. The recommendations are described below.
- lines 84-85 in the introduction “The detection and quantification of yeast and mould by conventional methods can be time-consuming and laborious” - please give examples of the use of conventional methods and citations
- line 219 - after the dot, there is an unnecessary dash in subscript
- line 315 – the word “are” is in superscript
- lines 322 – 336 – “is associated” is repeated too many times
In the manuscript, the authors describe the application of the NIR spectroscopy. Please provide selected NIR spectra and add a table with functional group assignments based on NIR spectra in order to facilitate a more straightforward presentation of the bands and their characteristics. This can be placed in the Materials and Methods section.
I suggested performing classification models based on partial least squares discriminant analysis (PLS-DA). Supervised classification methods can be used for predictive and descriptive modelling for a high-dimensional spectroscopic dataset. Therefore, the authors obtain better discrimination of the NIR spectra of analysed samples. On the other hand, I will understand that there is little time to do additional analysis.
Minor editing of English language can be applied.
Author Response
Answers Reviewer 2
- lines 84-85 in the introduction “The detection and quantification of yeast and mould by conventional methods can be time-consuming and laborious” - please give examples of the use of conventional methods and citations. We have added the examples requested by the reviewer.
- line 219 - after the dot, there is an unnecessary dash in subscript. We tried to removed from the revised version, however, still appeared in the document.
- line 315 – the word “are” is in superscript. We have corrected as suggested by the reviewer.
- lines 322 – 336 – “is associated” is repeated too many times. We have reworded the section and removing the “associated” as suggested by the reviewer.
In the manuscript, the authors describe the application of the NIR spectroscopy. Please provide selected NIR spectra and add a table with functional group assignments based on NIR spectra in order to facilitate a more straightforward presentation of the bands and their characteristics. This can be placed in the Materials and Methods section. This information is also provided in the description of the loadings (see lines 332 to 336).
I suggested performing classification models based on partial least squares discriminant analysis (PLS-DA). Supervised classification methods can be used for predictive and descriptive modelling for a high-dimensional spectroscopic dataset. Therefore, the authors obtain better discrimination of the NIR spectra of analysed samples. On the other hand, I will understand that there is little time to do additional analysis. We thank the reviewer for this comment and suggestion. However, in this case, we have better results using this approach. We did not have enough samples to attempt a classification as there were other sources of variability confounding the results (different wastes, etc). In addition for some of the samples, not enough samples to carried out a classification model.
Reviewer 3 Report
Line 240 and 241, use et al. instead of all the surnnames of the researchers.
Consider removing the exterior lines of the Figures.
Tables should only have three horizontal lines.
Overall, the standard deviations shown in Table 2 are too high. The Rcv2 values from Table 3 are too low to be accurately used for YMC prediction, even for screening purposes.
A control is missing in the research. A "common" feed should be used as control. Also, the time of the disposal and storage of the wastes used as feed should be at least indicated, since it is another important factor, as well as the temperature.
Line 322-336, this discussion should focus on the functional groups that some yeasts and moulds could have, thus presenting a more profound comparison.
The conclusions should be rewritten with a more realistica approach, since they are too "optimistic".
If possible, include PLS algorithm for further explanation and some representative model.
Author Response
Answers Reviewer 3
Line 240 and 241, use et al. instead of all the surnnames of the researchers. We have corrected as suggested by the reviewer.
Consider removing the exterior lines of the Figures. We could not remove the lines as they were imported from the software, sorry for this.
Tables should only have three horizontal lines. We have corrected and used only three horizontal lines as suggested by the reviewer.
Overall, the standard deviations shown in Table 2 are too high. The Rcv2 values from Table 3 are too low to be accurately used for YMC prediction, even for screening purposes. The high SD is due to the intrinsic characteristics of the routine method used. The method is based on the visual counting of the colonies for a given mould or yeast. Considering the number of samples and the method used the RPD are in the range defined by other authors as good for screening purposes of these type of samples and diseases. This variability is also related with the highly variability between waste samples, etc.
A control is missing in the research. A "common" feed should be used as control. Also, the time of the disposal and storage of the wastes used as feed should be at least indicated, since it is another important factor, as well as the temperature. We thank the comment from the reviewer. However, the experiments and samples were developed in a real industrial setting. It is almost impossible to define a “control” sample as the food waste can highly varies among days. The purposes of these experiments was to collect real samples and not create artificial diets. The samples, temperature, disposal and storage of the waste was also varied as these experiments and sampling were carried out by the industry following the protocols and regulations stipulated by the authorities.
Line 322-336, this discussion should focus on the functional groups that some yeasts and moulds could have, thus presenting a more profound comparison. The PLS models were based on the spectra of the larvae, feed, etc combined with the reference method (counting). The purpose of development the models was to measure the presence of yeast and moulds in the whole samples, not to characterise the yeast and mould alone. Therefore, it is not the aim of the paper to characterise the mould or yeast as proposed by the reviewer.
The conclusions should be rewritten with a more realistica approach, since they are too "optimistic". We considered that the conclusion reflects the finding of this experiment based on the use of NIR spectroscopy combined with chemometrics as in many other reports found in the scientific literature.
If possible, include PLS algorithm for further explanation and some representative model. The PLS algorithm is not easy to present. Usually, the report of PLS models is based on the presentation of the statistic and the interpretation of the loadings as in many other reports found in the scientific literature.
Round 2
Reviewer 1 Report
The raw spectra shown in the authors' reply have good quality and high variance. But I still insist on showing them in the manuscript or in supplementary material, because it is important for readers.
I also insist on an adequate discussion of the model complexity in the paper. RMSEC and RMSECV versus nLV dependencies should be presented. The loadings you show for the optimal models are not convincing that the model is healthy, because they are based on a smoothed derivative and are quite noisy, anyway. The model with 10 LV is an overfitting, because 75 samples cannot adequately cover the 10-dimensional factor space. At least 2^n samples are required. I assume that you have a long plateau in your RMSE/nLV function after n>5 or 6. Therefore, you can easily sacrifice the excessive model complexity at the expense of somewhat lower R2, which probably would not be critical for your final results.
My suggestion is to show the frass model statistics for 5 or 6 LVs, because you cannot justify the need for higher complexity.
minor proofreading is necessary
Author Response
I want to start telling the Editor that is very upsetting for the corresponding author that reviewer 1 has implied or assumed (using the reviewer’s word) that the quality of the NIR spectra was not good and the models were overfitted. The corresponding author has more than 30 years of experience in the applications of NIR spectroscopy, have delivered several workshops on the use of NIR and chemometrics, has organised local and international conferences in the field of NIR spectroscopy, has received the Tomas Hirschfeld Award from the International Council of Near Infrared Spectroscopy for his outstanding contributions in the field of NIR spectroscopy, has published more than 500 papers in the field of chemometrics and infrared spectroscopy and is the Editor of serval international Journals and never receive this kind of comments. The corresponding author also believes in the quality, high standards, knowledge and objectivity needed in the reviewing process. Therefore, the answers for reviewer 1 comments are below.
- The raw spectra shown in the authors' reply have good quality and high variance. But I still insist on showing them in the manuscript or in supplementary material, because it is important for readers. In the experience of the corresponding author and for the solely purpose of this study, showing the raw spectra does not add any more relevant information. We do not agree with the comments from the reviewer 1 implying the poor quality of the NIR spectra collected in this study. What are the elements considered by the reviewer to judge the quality of the spectra? Is the reviewer only considered the variance of the spectra? I have assumed that the reviewer 1 will know that any pre-processing will modify or improve any of these issues (e.g. spikes, noise, etc). I am also assuming that the reviewer 1 will know that before the utilization of chemometrics, the spectra must be analysed, and some level of pre-processing will be required. Nevertheless, in respect of the reviewing process, we have added the raw spectra, baseline correction, first derivative and second derivative of all the samples as supplementary file as requested by the reviewer.
- I also insist on an adequate discussion of the model complexity in the paper. RMSEC and RMSECV versus nLV dependencies should be presented. The loadings you show for the optimal models are not convincing that the model is healthy, because they are based on a smoothed derivative and are quite noisy, anyway. We do not agree with this statement. Reviewer 1 seems to not understand or ignore the use of pre-processing and its relevance in NIR spectroscopy. Some details and reference are provided in the following sections. The use of smoothing and derivatives aim to enhance the quality of the data by improving the signal-to-noise (S/N) ratio, reducing unwanted sources of variation, and correcting systematic errors in the data. The Savitzky-Golay (S-G) smoothing is applied to mitigate the effect of noise from the spectrum and is used to eliminate small variations in the absorbance or peaks which are unlikely to have a significant effect during data mining and interpretation. Smoothing aims to reduce high-frequency noise, such as removing "spikes" in the data. However, one important limitation to consider during the application of smoothing, is that this process might also remove high frequency components that represent useful information. As in many cases with the utilization of pre-processing techniques the trade-off between the visual appearance and the relevant information must be considered when selecting the appropriate smoothing window size. In addition to smoothing, derivatives were used in this study as they provide with an effective improvement in signal resolution, allowing to separate two or more components that could have overlapping spectra. Derivatives might provide discrimination in favour of the sharpest features of a spectrum, used to eliminate interferences by broad band constituents. During the process of derivatization, the polynomial order is decreased by one for each derivative calculation. For example, the first derivative eliminates a constant offset while the second derivative removes both the offset and the linear term (e.g. linear fitting applied to the spectrum), and so forth (e.g. third, fourth derivatives). Variations in particle size can also contribute to changes in the offset and slope of the spectra (e.g. NIR spectra). For example, in NIR spectroscopy the first-order derivative is the rate of change of absorbance with respect to the wavelength. The second order derivative in particular can accentuate sharp spectral features and resolve overlapping bands. Furthermore, the utilization of derivatives in spectroscopy can effectively reduce the effects of scattering by eliminating additive offsets which are independent of the wavelength (first derivatives) or change linearly with wavelength by removing offsets as in the case of using the second derivative. Of course, after pre-processing care must be taken in the interpretation of the loadings or coefficients of regression. Some references for the reviewer to consider:
Savitzky, A., Golay, M.J.E., 1964. Smoothing and differentiation of data by simplified least squares procedures. Anal. Chem. 36 (8), 1627–1639.
Duhamel, P.; Vetterli. M. (1990). Fast Fourier transformations; a tutorial review and state of the art. Signal Processing 19: 259-299.
Rinnan, A., 2014. Pre-processing in vibrational spectroscopy—when, why and how. Anal. Methods 6, 7124–7129.
Wetzel, D.L.B., 1998. Analytical near infrared spectroscopy. In: Wetzel, D.L.B., Charalambous, G. (Eds.), Instrumental Methods in Food and Beverage Analysis. Elsevier, Amsterdam, pp. 141–194.
Katsumoto, Y., Jiang, J.-H., Berry, R.J., Ozaki, Y., 2001. Modern pre-treatment methods in NIR spectroscopy. Near Infrared Anal. 2, 29–36.
Manley, M.; Baeten, V (2018). Spectroscopic Technique: Near Infrared (NIR) Spectroscopy Modern Techniques for Food Authentication.
Wetzel, D.L., 1983. Near infrared reflectance analysis of major components in foods. In: Charalambous, G., Inglett, G. (Eds.), Instrumental Analysis of Foods—Recent Progress. In: vol. 1. Academic Press, London, pp. 183–202.
Duckworth, J. Mathematical Data Pre-processing. Book Editor(s):Craig A. Roberts, Jerry Workman Jr., James B. Reeves III. Agronomy Monographs. 2004
Naes, T.; Isaksson, T.; Fearn, T.; Davies, T. (2002). A User-friendly guide to multivariate calibration and classification. NIR Publications, Chichester, UK.
Geladi, P., MacDougall, D., Martens, H., 1985. Linearization and scatter-correction for near infrared reflectance spectra of meat. Appl. Spectrosc. 39 (3), 491–500.
- The model with 10 LV is an overfitting, because 75 samples cannot adequately cover the 10-dimensional factor space. At least 2^n samples are required. We do not agree with this suggestion. First, the development of a calibration is not mathematical or statistical exercise. The use of chemometrics is considered an optimization process. The software used in this study (as many other software packages) has different measures that minimise or avoid the overfitting or underfitting of the model. During the process, the software used the PRESS function (Predicted Residual Error Sum of Squares). The PRESS function is the residual Y variance over the number of validation objects or samples. PRESS is often used to assess whether an individual, new component represents a significant addition to the model or not. The other assumption made by the anonymous reviewer that is wrong, is about the number of LV, factors or components. The largest number of components is either n-1 (number of samples – 1) or p (number of variables) depending which is the smaller. For example, in this study [75 samples / spectra x 464 variables (wavelengths)] the maximum number of LV, factors or components is 74. This is because the larger number of LV, factors, etc is limited by the number of samples. Extensive literature exists, but good books in chemometrics explain this or even the manual of the software used. Because the PRESS function, the use of cross validation and the definitions provided above, none of these models are overfitted as assumed by the anonymous reviewer.
- I assume that you have a long plateau in your RMSE/nLV function after n>5 or 6. Therefore, you can easily sacrifice the excessive model complexity at the expense of somewhat lower R2, which probably would not be critical for your final results. We do not agree with this suggestion. First, the development of a calibration is not mathematical or statistical exercise. The use of chemometrics is considered an optimization process. The results reported here came from an optimization process not from a trial error exercise, or by plotting the error vs the number of LV and select the one that the user like the most. The software used in this study (as many other software packages) has different measures that select the optimal number of LV, factors or components that avoid the overfitting and underfitting of the model. As explained above, during the process, the software used the PRESS function (Predicted Residual Error Sum of Squares). The PRESS function is the residual Y variance over the number of validation objects or samples. PRESS is often used to assess whether an individual, new component represents a significant addition to the model. The other issue that is wrong, is about the number of LV, factors or components. The largest number of components is either n-1 (number of samples – 1) or p (number of variables) depending on which is the smaller. For example, in this study 75 samples / spectra x 464 variables (wavelengths) the maximum number of LV, factors or components is 74. This is because the larger number of LV, factors, etc is limited by the number of samples. Extensive literature exists, but good books in chemometrics explain this or the manual of the software used. Because the PRESS function, the use of cross validation and the definition provided above, none of these models are overfitted or underfitted as assumed by the anonymous reviewer.
- My suggestion is to show the frass model statistics for 5 or 6 LVs, because you cannot justify the need for higher complexity. We do not agree with this suggestion. I will like to know what is the definition of complexity or even what does the reviewer define as complexity. Please also note that complexity and variability define different issues in NIR spectroscopy. Once again, I will reiterate that the development of a calibration is not mathematical or statistical exercise. The use of chemometrics is considered an optimization process. The PRESS function, the use of cross validation and the definition provided above, none of these models are overfitted as assumed by the reviewer 1.
Reviewer 3 Report
Table 1 should have the same format as the others (only 3 horizontal lines).
It was previously mentioned that the time of the disposal and storage of the wastes used as feed should be at least indicated, since it is another important factor, as well as the temperature. The response was: "The samples, temperature, disposal and storage of the waste was also varied as these experiments and sampling were carried out by the industry following the protocols and regulations stipulated by the authorities.". If not even temperatures were recorded, how would it be possible to recreate this study? Also, "the protocols and regulations stipulated by the authorities" were not included and should be properly referenced. I realize that this study was developed in a real industry setting, nevertheless, some factors ought have been monitored.
The conclusions should indicate that this study might be used to assess YMC. This writing is not consistent with the abstract, since in the abstract the conclusions seem more cautious.
Author Response
Table 1 should have the same format as the others (only 3 horizontal lines). We have reformatted the table as suggested by the reviewer.
It was previously mentioned that the time of the disposal and storage of the wastes used as feed should be at least indicated, since it is another important factor, as well as the temperature. The response was: "The samples, temperature, disposal and storage of the waste was also varied as these experiments and sampling were carried out by the industry following the protocols and regulations stipulated by the authorities.". If not even temperatures were recorded, how would it be possible to recreate this study? Also, "the protocols and regulations stipulated by the authorities" were not included and should be properly referenced. I realize that this study was developed in a real industry setting, nevertheless, some factors ought have been monitored. The temperature and other conditions were monitored by each of the facilities. The feeding, waste management system, rearing conditions, are protected by industrial rights. Overall, this study has used industrial operating procedures and standards that can be used by similar type of industries worldwide. We have added these sentences in the materials and method section.
The conclusions should indicate that this study might be used to assess YMC. This writing is not consistent with the abstract, since in the abstract the conclusions seem more cautious. We have reworded as suggested by the reviewer.